# Efficient Streaming Language Models with Attention Sinks

**Guangxuan Xiao**[1*]  **Yuandong Tian**[2]  **Beidi Chen**[3]  **Song Han**[1,4]  **Mike Lewis**[2]

[1] Massachusetts Institute of Technology    [2] Meta AI
[3] Carnegie Mellon University    [4] NVIDIA
`https://github.com/mit-han-lab/streaming-llm`

## Abstract

Deploying Large Language Models (LLMs) in streaming applications such as multi-round dialogue, where long interactions are expected, is urgently needed but poses two major challenges. Firstly, during the decoding stage, caching previous tokens' Key and Value states (KV) consumes extensive memory. Secondly, popular LLMs cannot generalize to longer texts than the training sequence length. Window attention, where only the most recent KVs are cached, is a natural approach — but we show that it fails when the text length surpasses the cache size. We observe an interesting phenomenon, namely *attention sink*, that keeping the KV of initial tokens will largely recover the performance of window attention. In this paper, we first demonstrate that the emergence of *attention sink* is due to the strong attention scores towards initial tokens as a "sink" even if they are not semantically important. Based on the above analysis, we introduce StreamingLLM, an efficient framework that enables LLMs trained with a *finite length* attention window to generalize to *infinite sequence length* without any fine-tuning. We show that StreamingLLM can enable Llama-2, MPT, Falcon, and Pythia to perform stable and efficient language modeling with up to 4 million tokens and more. In addition, we discover that adding a placeholder token as a dedicated attention sink during pre-training can further improve streaming deployment. In streaming settings, StreamingLLM outperforms the sliding window recomputation baseline by up to 22.2× speedup. Code and datasets are provided in the link.

## 1 Introduction

Large Language Models (LLMs) (Radford et al., 2018; Brown et al., 2020; Zhang et al., 2022; OpenAI, 2023; Touvron et al., 2023a;b) are becoming ubiquitous, powering many natural language processing applications such as dialog systems (Schulman et al., 2022; Taori et al., 2023; Chiang et al., 2023), document summarization (Goyal & Durrett, 2020; Zhang et al., 2023a), code completion (Chen et al., 2021; Rozière et al., 2023) and question answering (Kamalloo et al., 2023). To unleash the full potential of pretrained LLMs, they should be able to efficiently and accurately perform long sequence generation. For example, an ideal ChatBot assistant can stably work over the content of recent day-long conversations. However, it is very challenging for LLM to generalize to longer sequence lengths than they have been pretrained on, e.g., 4K for Llama-2 Touvron et al. (2023b).

The reason is that LLMs are constrained by the attention window during pre-training. Despite substantial efforts to expand this window size (Chen et al., 2023; kaiokendev, 2023; Peng et al., 2023) and improve training (Dao et al., 2022; Dao, 2023) and inference (Pope et al., 2022; Xiao et al., 2023; Anagnostidis et al., 2023; Wang et al., 2021; Zhang et al., 2023b) efficiency for lengthy inputs, the acceptable sequence length remains intrinsically *finite*, which doesn't allow persistent deployments.

In this paper, we first introduce the concept of LLM streaming applications and ask the question:

*Can we deploy an LLM for infinite-length inputs without sacrificing efficiency and performance?*

---

*Part of the work done during an internship at Meta AI.

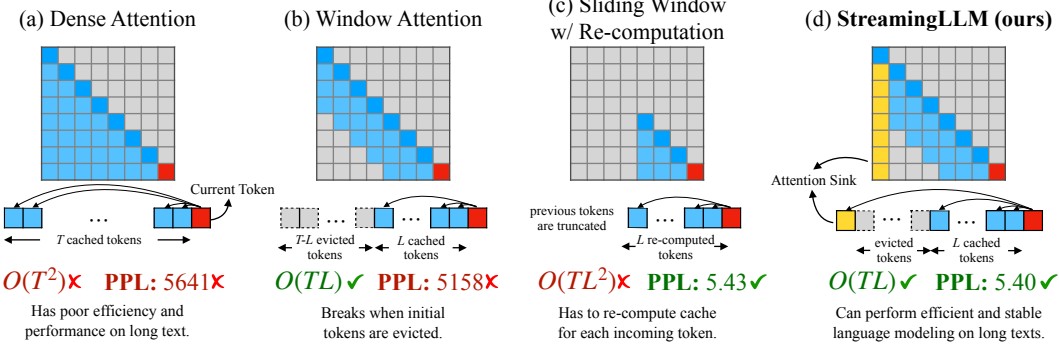

Figure 1: **Illustration of StreamingLLM *vs*. existing methods.** The language model, pre-trained on texts of length $L$, predicts the $T$th token ($T \gg L$). (a) Dense Attention has $O(T^2)$ time complexity and an increasing cache size. Its performance decreases when the text length exceeds the pre-training text length. (b) Window Attention caches the most recent $L$ tokens' KV. While efficient in inference, performance declines sharply once the starting tokens' keys and values are evicted. (c) Sliding Window with Re-computation rebuilds the KV states from the $L$ recent tokens for each new token. While it performs well on long texts, its $O(TL^2)$ complexity, stemming from quadratic attention in context re-computation, makes it considerably slow. (d) StreamingLLM keeps the *attention sink* (several initial tokens) for stable attention computation, combined with the recent tokens. It's efficient and offers stable performance on extended texts. Perplexities are measured using the Llama-2-13B model on the first book (65K tokens) in the PG-19 test set.

When applying LLMs for infinite input streams, two primary challenges arise:

1. During the decoding stage, Transformer-based LLMs cache the Key and Value states (KV) of all previous tokens, as illustrated in Figure 1 (a), which can lead to excessive memory usage and increasing decoding latency (Pope et al., 2022).

2. Existing models have limited length extrapolation abilities, i.e., their performance degrades (Press et al., 2022; Chen et al., 2023) when the sequence length goes beyond the attention window size set during pre-training.

An intuitive approach, known as window attention (Beltagy et al., 2020) (Figure 1 b), maintains only a fixed-size sliding window on the KV states of most recent tokens. Although it ensures constant memory usage and decoding speed after the cache is initially filled, the model collapses once the sequence length exceeds the cache size, i.e., *even just evicting the KV of the first token*, as illustrated in Figure 3. Another strategy is the sliding window with re-computation (shown in Figure 1 c), which rebuilds the KV states of recent tokens for each generated token. While it offers strong performance, this approach is significantly slower due to the computation of quadratic attention within its window, making this method impractical for real-world streaming applications.

To understand the failure of window attention, we find an interesting phenomenon of autoregressive LLMs: a surprisingly large amount of attention score is allocated to the initial tokens, irrespective of their relevance to the language modeling task, as visualized in Figure 2. We term these tokens "**attention sinks**". Despite their lack of semantic significance, they collect significant attention scores. We attribute the reason to the Softmax operation, which requires attention scores to sum up to one for all contextual tokens. Thus, even when the current query does not have a strong match in many previous tokens, the model still needs to allocate these unneeded attention values somewhere so it sums up to one. The reason behind *initial* tokens as sink tokens is intuitive: initial tokens are visible to almost all subsequent tokens because of the autoregressive language modeling nature, making them more readily trained to serve as attention sinks.

Based on the above insights, we propose StreamingLLM, a simple and efficient framework that enables LLMs trained with a finite attention window to work on text of infinite length without fine-tuning. StreamingLLM exploits the fact that attention sinks have high attention values, and preserving them can maintain the attention score distribution close to normal. Therefore, StreamingLLM simply keeps the attention sink tokens' KV (with just 4 initial tokens sufficing) together with the sliding window's KV to anchor the attention computation and stabilize the model's performance. With StreamingLLM, models including Llama-2-[7, 13, 70]B, MPT-[7, 30]B, Falcon-[7, 40]B, and Pythia-[2.9,6.9,12]B can reliably model 4 million tokens, and potentially even more. Compared with the only viable baseline, sliding window with recomputation, StreamingLLM achieves up to 22.2× speedup, realizing the streaming use of LLMs.

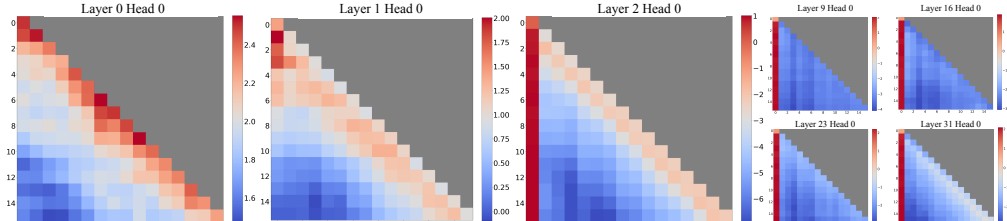

Figure 2: Visualization of the *average* attention logits in Llama-2-7B over 256 sentences, each with a length of 16. Observations include: (1) The attention maps in the first two layers (layers 0 and 1) exhibit the "local" pattern, with recent tokens receiving more attention. (2) Beyond the bottom two layers, the model heavily attends to the initial token across all layers and heads.

Furthermore, we confirm our attention sink hypothesis and demonstrate that language models can be pre-trained to require only a single attention sink token for streaming deployment. Specifically, we suggest that an extra learnable token at the beginning of all training samples can serve as a designated attention sink. By pre-training 160-million parameter language models from scratch, we demonstrate that adding this single sink token preserves the model's performance in streaming cases. This stands in contrast to vanilla models, which necessitate the reintroduction of multiple initial tokens as attention sinks to achieve the same performance level.

Finally, we emphasize that StreamingLLM efficiently generates coherent text from tokens within the KV cache without extending the LLMs' context length. It suits continuous operation needs with minimal memory use and past data reliance. Additionally, StreamingLLM can complement context extension methods to increase the attendable recent context.

## 2 RELATED WORK

Extensive research has been done on applying LLMs to lengthy texts, with three main areas of focus: **Length Extrapolation**, **Context Window Extension**, and **Improving LLMs' Utilization of Long Text**. While seemingly related, it's worth noting that progress in one direction doesn't necessarily lead to progress in the other. For example, extending the context size of LLMs doesn't improve the model's performance beyond the context size, and neither approach ensures effective use of the long context. Our StreamingLLM framework primarily lies in the first category, where LLMs are applied to text significantly exceeding the pre-training window size, potentially even of infinite length. We do not expand the attention window size of LLMs or enhance the model's memory and usage on long texts. The last two categories are orthogonal to our focus and could be integrated with our techniques.

**Length extrapolation** aims to enable language models trained on shorter texts to handle longer ones during testing. A predominant avenue of research targets the development of relative position encoding methods for Transformer models, enabling them to function beyond their training window. One such initiative is Rotary Position Embeddings (RoPE) (Su et al., 2021), which transforms the queries and keys in every attention layer for relative position integration. Despite its promise, subsequent research (Press et al., 2022; Chen et al., 2023) indicated its underperformance on text that exceeds the training window. Another approach, ALiBi (Press et al., 2022), biases the query-key attention scores based on their distance, thereby introducing relative positional information. While this exhibited improved extrapolation, our tests on MPT models highlighted a breakdown when the text length was vastly greater than the training length. Current methodologies, however, have yet to achieve infinite length extrapolation, causing no existing LLMs to fit for streaming applications.

**Context Window Extension** centers on expanding the LLMs' context window, enabling the processing of more tokens in one forward pass. A primary line of work addresses the training efficiency problem. Given the attention to computation's quadratic complexity during training, developing a long-context LLM is both a computational and memory challenge. Solutions have ranged from system-focused optimizations like FlashAttention (Dao et al., 2022; Dao, 2023), which accelerates attention computation and reduces memory footprint, to approximate attention methods (Zaheer et al., 2020b; Beltagy et al., 2020; Wang et al., 2020; Kitaev et al., 2020) that trade model quality for efficiency. Recently, there has been a surge of work on extending pre-trained LLMs with RoPE (Chen et al., 2023; kaiokendev, 2023; bloc97, 2023; Peng et al., 2023), involving position interpolation and fine-tuning. However, all the aforementioned techniques only extend LLMs' context window to a limited extent, which falls short of our paper's primary concern of handling limitless inputs.

Figure 3: Language modeling perplexity on texts with 20K tokens across various LLM. Observations reveal consistent trends: (1) Dense attention fails once the input length surpasses the pre-training attention window size. (2) Window attention collapses once the input length exceeds the cache size, i.e., the initial tokens are evicted. (3) StreamingLLM demonstrates stable performance, with its perplexity nearly matching that of the sliding window with re-computation baseline.

**Improving LLMs' Utilization of Long Text** optimizes LLMs to better capture and employ the content within the context rather than merely taking them as inputs. As highlighted by Liu et al. and Li et al., success in the previously mentioned two directions does not necessarily translate to competent utilization of lengthy contexts. Addressing this effective usage of prolonged contexts within LLMs is still a challenge. Our work concentrates on stably harnessing the most recent tokens, enabling the seamless streaming application of LLMs.

## 3 STREAMINGLLM

### 3.1 THE FAILURE OF WINDOW ATTENTION AND ATTENTION SINKS

While the window attention technique offers efficiency during inference, it results in an exceedingly high language modeling perplexity. Consequently, the model's performance is unsuitable for deployment in streaming applications. In this section, we use the concept of *attention sink* to explain the failure of window attention, serving as the inspiration behind StreamingLLM.

**Identifying the Point of Perplexity Surge.** Figure 3 shows the perplexity of language modeling on a 20K token text. It is evident that perplexity spikes when the text length surpasses the cache size, led by the exclusion of initial tokens. This suggests that the initial tokens, regardless of their distance from the predicted tokens, are crucial for maintaining the stability of LLMs.

**Why do LLMs break when removing *initial* tokens' KV?** We visualize attention maps from all layers and heads of the Llama-2-7B and models in Figure 2. We find that, beyond the bottom two layers, the model consistently focuses on the initial tokens across all layers and heads. The implication is clear: removing these initial tokens' KV will remove a considerable portion of the denominator in the SoftMax function (Equation 1) in attention computation. This alteration leads to a significant shift in the distribution of attention scores away from what would be expected in normal inference settings.

$$\text{SoftMax}(x)_i = \frac{e^{x_i}}{e^{x_1} + \sum_{j=2}^{N} e^{x_j}}, \quad x_1 \gg x_j, j \in 2, \dots, N \tag{1}$$

There are two possible explanations for the importance of the initial tokens in language modeling: (1) Either their semantics are crucial, or (2) the model learns a bias towards their absolute position. To distinguish between these possibilities, we conduct experiments (Table 1), wherein the first four tokens are substituted with the linebreak token "\n". The observations indicate that the model still significantly emphasizes these initial linebreak tokens. Furthermore, reintroducing them restores the language modeling perplexity to levels comparable to having the original initial tokens. This suggests that the absolute position of the starting tokens, rather than their semantic value, holds greater significance.

**LLMs attend to Initial Tokens as Attention Sinks.** To explain why the model disproportionately focuses on initial tokens—regardless of their semantic relevance to language modeling, we introduce the concept of "*attention sink*". The nature of the SoftMax function (Equation 1) prevents all attended tokens from having zero values. This requires aggregating some information from other tokens across all heads in all layers, even if the current embedding has sufficient self-contained information for its prediction. Consequently, the model tends to dump unnecessary attention values to specific tokens. A similar observation has been made in the realm of quantization outliers (Xiao et al., 2023; Bondarenko et al., 2023), leading to the proposal of SoftMax-Off-by-One (Miller, 2023) as a potential remedy.

Table 1: Window attention has poor performance on long text. The perplexity is restored when we reintroduce the initial four tokens alongside the recent 1020 tokens (4+1020). Substituting the original four initial tokens with linebreak tokens "\n" (4"\n"+1020) achieves comparable perplexity restoration. Cache config x+y denotes adding x initial tokens with y recent tokens. Perplexities are measured on the first book (65K tokens) in the PG19 test set.

| Llama-2-13B | PPL (↓) |
|---|---|
| 0 + 1024 (Window) | 5158.07 |
| 4 + 1020 | 5.40 |
| 4"\n"+1020 | 5.60 |

Table 2: Effects of reintroduced initial token numbers on StreamingLLM. (1) Window attention (0+y) has a drastic increase in perplexity. (2) Introducing one or two initial tokens doesn't fully restore model perplexity, showing that the model doesn't solely use the first token as the attention sink. (3) Introducing four initial tokens generally suffices; further additions have diminishing returns. Cache config x+y denotes adding x initial tokens to y recent tokens. Perplexities are evaluated on 400K tokens in the concatenated PG19 test set.

| Cache Config | 0+2048 | 1+2047 | 2+2046 | 4+2044 | 8+2040 |
|---|---|---|---|---|---|
| Falcon-7B | 17.90 | 12.12 | 12.12 | 12.12 | 12.12 |
| MPT-7B | 460.29 | 14.99 | 15.00 | 14.99 | 14.98 |
| Pythia-12B | 21.62 | 11.95 | 12.09 | 12.09 | 12.02 |

| Cache Config | 0+4096 | 1+4095 | 2+4094 | 4+4092 | 8+4088 |
|---|---|---|---|---|---|
| Llama-2-7B | 3359.95 | 11.88 | 10.51 | 9.59 | 9.54 |

Why do various autoregressive LLMs, such as Llama-2, MPT, Falcon, and Pythia, consistently focus on *initial tokens* as their attention sinks, rather than other tokens? Our explanation is straightforward: Due to the sequential nature of autoregressive language modeling, initial tokens are visible to all subsequent tokens, while later tokens are only visible to a limited set of subsequent tokens. As a result, initial tokens are more easily trained to serve as attention sinks, capturing unnecessary attention.

We've noted that LLMs are typically trained to utilize multiple initial tokens as attention sinks rather than just one. As illustrated in Figure 2, the introduction of four initial tokens, as attention sinks, suffices to restore the LLM's performance. In contrast, adding just one or two doesn't achieve full recovery. We believe this pattern emerges because these models didn't include a consistent starting token across all input samples during pre-training. Although Llama-2 does prefix each paragraph with a "" token, it's applied before text chunking, resulting in a mostly random token occupying the zeroth position. This lack of a uniform starting token leads the model to use several initial tokens as attention sinks. We hypothesize that by incorporating a stable learnable token at the start of all training samples, it could singularly act as a committed attention sink, eliminating the need for multiple initial tokens to ensure consistent streaming. We will validate this hypothesis in Section 3.3.

## 3.2 ROLLING KV CACHE WITH ATTENTION SINKS

To enable LLM streaming in already trained LLMs, we propose a straightforward method that can recover window attention's perplexity without any model finetuning. Alongside the current sliding window tokens, we reintroduce a few starting tokens' KV in the attention computation.

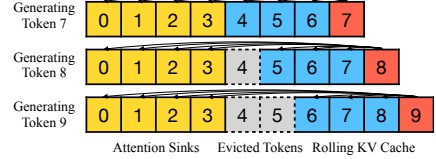

Figure 4: The KV cache of StreamingLLM.

The KV cache in StreamingLLM can be conceptually divided into two parts, as illustrated in Figure 4: (1) Attention sinks (four initial tokens) stabilize the attention computation; 2) Rolling KV Cache retains the most recent tokens, crucial for language modeling. StreamingLLM' design is versatile and can be seamlessly incorporated into any autoregressive language model that employs relative positional encoding, such as RoPE (Su et al., 2021) and ALiBi (Press et al., 2022).

When determining the relative distance and adding positional information to tokens, StreamingLLM focuses on positions *within the cache* rather than those *in the original text*. This distinction is crucial for StreamingLLM's performance. For instance, if the current cache (Figure 4) has tokens [0, 1, 2, 3, 6, 7, 8] and is in the process of decoding the 9th token, the positions assigned are [0, 1, 2, 3, 4, 5, 6, 7], rather than the positions in the original text, which would be [0, 1, 2, 3, 6, 7, 8, 9].

For encoding like RoPE, we cache the Keys of tokens *prior to* introducing the rotary transformation. Then, we apply position transformation to the keys in the rolling cache at each decoding phase. On the other hand, integrating with ALiBi is more direct. Here, the contiguous linear bias is applied instead of a 'jumping' bias to the attention scores. This method of assigning positional embedding within the cache is crucial to StreamingLLM's functionality, ensuring that the model operates efficiently even beyond its pre-training attention window size.

### 3.3 PRE-TRAINING LLMs WITH ATTENTION SINKS

As elaborated in Section 3.1, a significant reason for the model's excessive attention to multiple initial tokens is the absence of a designated sink token to offload excessive attention scores. Due to this, the model inadvertently uses globally visible tokens, primarily the initial ones, as attention sinks. A potential remedy can be the intentional inclusion of a global trainable attention sink token, denoted as a "Sink Token", which would serve as a repository for unnecessary attention scores. Alternatively, replacing the conventional SoftMax function with a variant like SoftMax-off-by-One (Miller, 2023),

$$\text{SoftMax}_1(x)_i = \frac{e^{x_i}}{1 + \sum_{j=1}^{N} e^{x_j}}, \qquad (2)$$

Table 3: Comparison of vanilla attention with prepending a zero token and a learnable sink token during pre-training. To ensure stable streaming perplexity, the vanilla model requires several initial tokens. While Zero Sink shows a slight improvement, it still needs other initial tokens. Conversely, the model trained with a learnable Sink Token shows stable streaming perplexity with only the sink token added. Cache config $x+y$ denotes adding $x$ initial tokens with $y$ recent tokens. Perplexity is evaluated on the first sample in the PG19 test set.

| Cache Config | 0+1024 | 1+1023 | 2+1022 | 4+1020 |
|---|---|---|---|---|
| Vanilla | 27.87 | 18.49 | 18.05 | 18.05 |
| Zero Sink | 29214 | 19.90 | 18.27 | 18.01 |
| Learnable Sink | 1235 | **18.01** | 18.01 | 18.02 |

which does not require the attention scores on all contextual tokens to sum up to one, may also be effective. Note that SoftMax$_1$ is equivalent to prepending a token with an all-zero Key and Value features in the attention computation. We denote this method as "Zero Sink" to fit our framework.

For validation, we pre-train three language models with 160 million parameters from scratch under identical settings. The first model utilizes the standard SoftMax attention (Vanilla), the second replaced the regular attention mechanism with SoftMax$_1$ (Zero Sink), and one prepending a learnable placeholder token (Sink Token) in all training samples. As shown in Table 3, while the zero sink alleviates the attention sink problem to some extent, the model still relies on other initial tokens as attention sinks. Introducing a sink token is highly effective in stabilizing the attention mechanism. Simply pairing this sink token with recent tokens sufficiently anchors the model's performance, and the resulting evaluation perplexity is even marginally improved. Given these findings, we recommend training future LLMs with a sink token in all samples to optimize streaming deployment.

## 4 EXPERIMENTS

We evaluate StreamingLLM using four prominent recent model families: Llama-2 (Touvron et al., 2023b), MPT (Team, 2023), PyThia (Biderman et al., 2023), and Falcon (Almazrouei et al., 2023). Notably, Llama-2, Falcon, and Pythia incorporate RoPE (Su et al., 2021), whereas MPT employs ALiBi (Press et al., 2022) — two of the most influential position encoding techniques in recent research. Our diverse model selection ensures the validity and robustness of our findings. We benchmark StreamingLLM against established baselines such as dense attention, window attention, and the sliding window approach with re-computation. In all subsequent experiments with StreamingLLM, we default to using four initial tokens as attention sinks unless stated otherwise.

### 4.1 LANGUAGE MODELING ON LONG TEXTS ACROSS LLM FAMILIES AND SCALES

We firstly evaluate StreamingLLM's language modeling perplexity using the concatenated PG19 (Rae et al., 2020) test set, which contains 100 long books. For Llama-2 models, the cache size is set at 2048, while for Falcon, Pythia, and MPT models, it's set at 1024. This is half the pre-training window size chosen to enhance visualization clarity.

Figure 3 illustrates that StreamingLLM can match the oracle baseline (sliding window with re-computation) in terms of perplexity on texts spanning 20K tokens. Meanwhile, the dense attention technique fails when the input length exceeds its pre-training window, and the window attention technique struggles when the input length surpasses the cache size, leading to the eviction of the initial tokens. In Figure 5, we further substantiate that StreamingLLM can reliably handle exceptionally extended texts, encompassing more than 4 million tokens, across a spectrum of model families and scales. This includes Llama-2-[7,13,70]B, Falcon-[7,40]B, Pythia-[2.8,6.9,12]B, and MPT-[7,30]B.

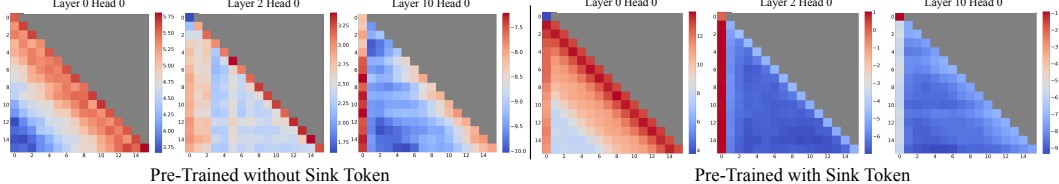

Figure 5: Language modeling perplexity of StreamingLLM on super long texts with 4 million tokens across various LLM families and scales. The perplexity remains stable throughout. We use the concatenated test set of PG19 (100 books) to perform language modeling, with perplexity fluctuations due to book transitions.

Figure 7: Visualization of average attention logits over 256 sentences, each 16 tokens long, comparing models pre-trained without (left) and with (right) a sink token. Both maps show the same layers and heads. Key observations: (1) Without a sink token, models show local attention in lower layers and increased attention to initial tokens in deeper layers. (2) With a sink token, there is clear attention directed at it across all layers, effectively collecting redundant attention. (3) With the presence of the sink token, less attention is given to other initial tokens, supporting the benefit of designating the sink token to enhance the streaming performance.

## 4.2 RESULTS OF PRE-TRAINING WITH A SINK TOKEN

To validate our suggestion that introducing a sink token to all pre-training samples improves streaming LLMs, we trained two language models, each with 160 million parameters, under identical conditions. While one model adhered to the original training settings, the other incorporated a sink token at the start of every training sample. Our experiments employed the Pythia-160M (Biderman et al., 2023) codebase and followed its training recipe. We train the models on an 8xA6000 NVIDIA GPU server using the deduplicated Pile (Gao et al., 2020) dataset. Apart from reducing the training batch size to 256, we retained all Pythia training configurations, including learning rate schedules, model initialization, and dataset permutations. Both models were trained for 143,000 steps.

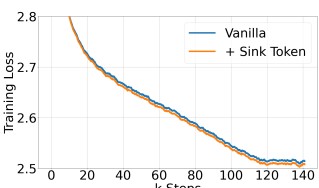

Figure 6: Pre-training loss curves of models w/ and w/o sink tokens. Two models have a similar convergence trend.

Table 4: Zero-shot accuracy (in %) across 7 NLP benchmarks, including ARC-[Challenge, Easy], HellaSwag, LAMBADA, OpenbookQA, PIQA, and Winogrande. The inclusion of a sink token during pre-training doesn't harm the model performance.

| Methods | ARC-c | ARC-e | HS | LBD | OBQA | PIQA | WG |
|---|---|---|---|---|---|---|---|
| Vanilla | 18.6 | 45.2 | 29.4 | 39.6 | 16.0 | 62.2 | 50.1 |
| +Sink Token | **19.6** | **45.6** | **29.8** | **39.9** | **16.6** | **62.6** | **50.8** |

**Convergence and Normal Model Performance.**    Including a sink token during pre-training has no negative impact on model convergence and subsequent performance on a range of NLP benchmarks. As depicted in Figure 6, models trained with a sink token exhibit similar convergence dynamics compared to their vanilla counterparts. We evaluate the two models on seven diverse NLP benchmarks, including ARC-[Challenge, Easy] (Clark et al., 2018), HellaSwag (Zellers et al., 2019), LAMBADA (Paperno et al., 2016), OpenbookQA (Mihaylov et al., 2018), PIQA (Bisk et al., 2020), and Winogrande (Sakaguchi et al., 2019). As shown in Table 4, the model pre-trained with a sink token performs similarly to that trained using the vanilla approach.

**Streaming Performance.**    As illustrated in Table 3, the streaming perplexities differ between models trained using traditional methods and those augmented with a sink token. Remarkably, the vanilla model requires the addition of multiple tokens as attention sinks to maintain stable streaming perplexity. In contrast, the model trained with a sink token achieves satisfactory streaming performance using just the sink token.

Table 5: Accuracy (in %) on the ARC-[Easy, Challenge] datasets. Questions were concatenated and answered in a streaming manner to mimic a real-world chat setting. The dense baseline fails due to Out-of-Memory (OOM) errors. Window attention has poor accuracy. StreamingLLM has comparable results with the one-shot sample-by-sample baseline. Window attention and StreamingLLM use cache sizes of 1024.

| Model | Llama-2-7B-Chat | | Llama-2-13B-Chat | | Llama-2-70B-Chat | |
|---|---|---|---|---|---|---|
| Dataset | Arc-E | Arc-C | Arc-E | Arc-C | Arc-E | Arc-C |
| One-shot | 71.25 | 53.16 | 78.16 | 63.31 | 91.29 | 78.50 |
| Dense | | | OOM | | | |
| Window | 3.58 | 1.39 | 0.25 | 0.34 | 0.12 | 0.32 |
| StreamingLLM | 71.34 | 55.03 | 80.89 | 65.61 | 91.37 | 80.20 |

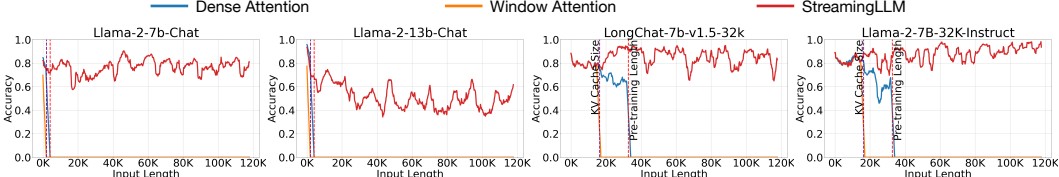

Figure 9: Performance on the StreamEval benchmark. Accuracies are averaged over 100 samples.

**Attention Visualization.** Figure 7 contrasts attention maps for models pre-trained with and without a sink token. The model without the sink token, similar to Llama-2-7B (Figure 2), shows early-layer local attention and deeper-layer focus on initial tokens. In contrast, models trained with a sink token consistently concentrate on the sink across layers and heads, indicating an effective attention offloading mechanism. This strong focus on the sink, with reduced attention to other initial tokens, explains the sink token's efficacy in enhancing model's streaming performance.

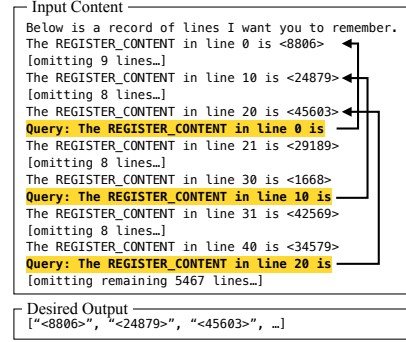

Figure 8: The first sample in StreamEval.

## 4.3 RESULTS ON STREAMING QUESTION ANSWERING WITH INSTRUCTION-TUNED MODELS

To show StreamingLLM's real-world applicability, we emulate multi-round question-answering using instruction-tuned LLMs, commonly used in real-world scenarios.

We first concatenate all question-answer pairs from the ARC-[Challenge, Easy] datasets, feed the continuous stream to Llama-2-[7,13,70]B-Chat models, and assess model completions at each answer position using an exact match criterion. As table 5 indicates, dense attention results in Out-of-Memory (OOM) errors, showing it unsuitable for this setting. While the window attention method works efficiently, it exhibits low accuracy due to random outputs when the input length exceeds the cache size. Conversely, StreamingLLM excels by efficiently handling the streaming format, aligning with the one-shot, sample-by-sample baseline accuracy.

Highlighting a more fitting scenario for StreamingLLM, we introduce a dataset, StreamEval, inspired by the LongEval (Li et al., 2023) benchmark. As depicted in Figure 8, diverging from LongEval's single query over a long-span setup, we query the model every 10 lines of new information. Each query's answer is consistently 20 lines prior, reflecting real-world instances where questions typically pertain to recent information. As illustrated in Figure 9, LLMs employing StreamingLLM maintain reasonable accuracy even as input lengths approach 120K tokens. In contrast, both dense and window attention fail at the pre-training text length and the KV cache size, respectively. Additionally, we utilize two context-extended models, LongChat-7b-v1.5-32k (Li et al., 2023) and Llama-2-7B-32K-Instruct (Together, 2023), to show that StreamingLLM can complement context extension techniques. Within StreamingLLM, context extension means broadening the maximum cache size of streaming LLMs, enabling the capture of broader local information.

Figure 10: Comparison of per-token decoding latency and memory usage between the sliding window approach with re-computation baseline and StreamingLLM, plotted against the cache size (attention window size) on the X-axis. StreamingLLM delivers a remarkable speedup of up to 22.2× per token and retains a memory footprint similar to the re-computation baseline.

## 4.4 ABLATION STUDIES

**Numbers of Initial Tokens.** In Table 2, we ablate the effect of adding varying numbers of initial tokens with recent tokens on the streaming perplexity. The results show the insufficiency of introducing merely one or two initial tokens, whereas a threshold of four initial tokens appears enough, with subsequent additions contributing marginal effects. This result justifies our choice of introducing 4 initial tokens as attention sinks in StreamingLLM.

**Cache Sizes.** In Table 6, we evaluate cache size's impact on StreamingLLM's perplexity. Contrary to intuition, increasing the cache size doesn't consistently lower the language modeling perplexity. This inconsistency shows a potential limitation where these models might not maximize the utility of the entire context they receive. Future research efforts should target enhancing these models' capabilities to utilize extensive contexts better.

## 4.5 EFFICENCY RESULTS

We benchmark StreamingLLM's decoding latency and memory usage against the sliding window with re-computation, which is the only baseline with acceptable quality. Both methods are implemented using the Huggingface Transformers library (Wolf et al., 2020) and tested on a single NVIDIA A6000 GPU using the Llama-2-7B and Llama-2-13B models. As shown in Figure 10, as the cache size increases, StreamingLLM's decoding speed has a linear growth. The sliding window with re-computation baseline has a quadratic rise in decoding latency. Thus, StreamingLLM achieves an impressive speedup, reaching up to 22.2× per token. Despite its reduced latency, StreamingLLM sustains a memory footprint consistent with the re-computation baseline.

Table 6: Effects of cache size on StreamingLLM's performance. Increasing the cache size in StreamingLLM doesn't consistently yield a decrease in perplexity, showing these models may not fully utilize the provided context. Cache config $x+y$ denotes adding $x$ initial tokens with $y$ recent tokens. Perplexity is evaluated on 400K tokens in the concatenated PG19 test set.

| Cache | 4+252 | 4+508 | 4+1020 | 4+2044 |
|---|---|---|---|---|
| Falcon-7B | 13.61 | 12.84 | **12.34** | 12.84 |
| MPT-7B | **14.12** | 14.25 | 14.33 | 14.99 |
| Pythia-12B | 13.17 | 12.52 | **12.08** | 12.09 |

| Cache | 4+508 | 4+1020 | 4+2044 | 4+4092 |
|---|---|---|---|---|
| Llama-2-7B | 9.73 | 9.32 | **9.08** | 9.59 |

## 5 CONCLUSION

Deploying LLMs in streaming applications is urgently needed but comes with challenges due to efficiency limitations and reduced performance with longer texts. Window attention provides a partial solution, but its performance plummets when initial tokens are excluded. Recognizing the role of these tokens as "attention sinks", we introduced StreamingLLM —a simple and efficient framework that enables LLMs to handle unlimited texts without fine-tuning. By adding attention sinks with recent tokens, StreamingLLM can efficiently model texts of up to 4 million tokens. We further show that pre-training models with a dedicated sink token can improve the streaming performance. StreamingLLM firstly decouples the LLM's pre-training window size and its actual text generation length, paving the way for the streaming deployment of LLMs.

REPRODUCIBILITY STATEMENT

All findings presented in this paper can be reproduced. We have made our code and datasets available in this github repo. The models used in this paper are all openly available, and we provide references to access them. Details regarding our experiments, including hyperparameters, training protocols, and evaluation methods, can be found in the Experiments section (Section 4). We are confident that with the provided resources, readers can reproduce the entirety of our presented results.

IMPACT STATEMENT

StreamingLLM has been widely adopted by various LLM serving solutions including NVIDIA TensorRT-LLM, Intel Extension for Transformers, HuggingFace Transformers, MLC LLM, etc.

ACKNOWLEDGEMENTS

This work is supported by MIT-IBM Watson AI Lab, Amazon and MIT Science Hub, National Science Foundation. We thank Angela Li for writing suggestions and demo making, Jingwei Zuo for proofreading, and Xiuyu Li for the suggestion on notations.

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

## A DISCUSSIONS

**Applications.** StreamingLLM is particularly suited for streaming applications, such as multi-round dialogues, where continuous operation without heavy reliance on extensive memory or historical data is crucial. For instance, in a daily assistant application based on LLMs, StreamingLLM enables the model to function seamlessly over extended periods. It bases its responses on recent interactions, thus avoiding the need for frequent cache refreshes. Traditional methods might require resetting the cache when the conversation length surpasses the training length, leading to a loss of recent context, or they might need to recompute key-value (KV) states from recent text history, which can be inefficient.

**Limitations.** While StreamingLLM improves the efficiency of LLMs in streaming contexts, it does not extend the models' context window or enhance their long-term memory capabilities. As detailed in Section C, the model is limited to operating within the confines of its current cache. Consequently, StreamingLLM is not suitable for tasks that demand long-term memory and extensive data dependency, such as long document question-answering (QA) and summarization. However, it excels in scenarios only requiring short-term memory, like daily conversations and short document QA, where its strength lies in generating coherent text from recent context without the need for cache refreshment.

**Broader Societal Impacts.** StreamingLLM significantly enhances the efficiency and accessibility of LLMs, democratizing their use across various sectors. By enabling nonstop and rapid interactions in applications like conversational agents, StreamingLLM improves user experiences, especially in scenarios requiring fixed-length models. This advancement allows for more seamless and contextually aware dialogues, potentially benefiting sectors like education, healthcare, and customer service. Additionally, StreamingLLM's efficiency in processing reduces the computational load, aligning with the need for environmentally sustainable AI technologies. This aspect is crucial in making advanced AI tools more accessible in regions with limited technological resources. However, the potential negative impacts of StreamingLLM mirror those associated with general language models, such as misinformation and biased content generation risks. It's essential to address these risks with robust ethical guidelines and safeguards. In summary, while StreamingLLM shares some risks common to language models, its positive contributions towards enhancing user experience, democratizing AI access, and promoting sustainability are noteworthy. These benefits underscore the importance of responsible deployment and ethical use of this technology.

## B ADDITIONAL RELATED WORKS

**Sparse Transformers.** The literature on efficient Transformer models primarily focuses on reducing the computational and memory complexity of the self-attention mechanism. A relevant line of work involves sparsifying the attention matrix by restricting the field of view to fixed, predefined patterns, such as local windows or block patterns with fixed strides (Tay et al., 2022). Sparse Transformer (Child et al., 2019) introduces sparse factorizations of the attention matrix, reducing the computational complexity of attention to $O(n\sqrt{n})$. LongFormer (Beltagy et al., 2020) combines dilated local windowed attention with task-motivated global attention. Extended Transformer Construction (ETC) Ainslie et al. (2020) presents a novel global-local attention mechanism, incorporating four types of attention patterns: global-to-global, local-to-local, local-to-global, and global-to-local. Building on ETC, BigBird (Zaheer et al., 2020a) proposes another linear complexity attention alternative, utilizing global tokens, local sliding window attentions, and random attention. However, these methods have several limitations. First, Sparse Transformer and ETC require custom GPU kernels for a specific block-sparse variant of matrix-matrix multiplication. Second, LongFormer, ETC, and BigBird all rely on a global attention pattern, which is unsuitable for autoregressive language models. Third, these methods are incompatible with pre-trained models, necessitating retraining from scratch. In contrast, our method offers ease of implementation using standard GPU kernels and is compatible with pre-trained autoregressive language models using dense attention, which are prevalent in the NLP community. This compatibility provides a significant advantage, allowing for the leveraging of existing pre-trained models without any fine-tuning.

**Concurrent Works.** Our research coincides with the work of Han et al., who conducted a theoretical study on the length generalization failure of language models, identifying three out-of-distribution factors. Their approach, inspired by this analysis, involves employing a "Λ"-shaped attention pattern

Table 7: Accuracy (in %) on StreamEval with increasing query-answer distance. Each line in StreamEval contains 23 tokens. Accuracies are averaged over 100 samples, and each sample contains 100 queries.

| Llama-2-7B-32K-Instruct | | Cache Config | | | |
|---|---|---|---|---|---|
| Line Distances | Token Distances | 4+2044 | 4+4092 | 4+8188 | 4+16380 |
| 20 | 460 | 85.80 | 84.60 | 81.15 | 77.65 |
| 40 | 920 | 80.35 | 83.80 | 81.25 | 77.50 |
| 60 | 1380 | 79.15 | 82.80 | 81.50 | 78.50 |
| 80 | 1840 | 75.30 | 77.15 | 76.40 | 73.80 |
| 100 | 2300 | 0.00 | 61.60 | 50.10 | 40.50 |
| 150 | 3450 | 0.00 | 68.20 | 58.30 | 38.45 |
| 200 | 4600 | 0.00 | 0.00 | 62.75 | 46.90 |
| 400 | 9200 | 0.00 | 0.00 | 0.00 | 45.70 |
| 600 | 13800 | 0.00 | 0.00 | 0.00 | 28.50 |
| 800 | 18400 | 0.00 | 0.00 | 0.00 | 0.00 |
| 1000 | 23000 | 0.00 | 0.00 | 0.00 | 0.00 |

and reconfiguring position encoding distances to enhance length generalization in LLMs. This approach bears a resemblance to our methodology. However, our work uncovers the "attention sink" phenomenon, wherein Transformer models tend to assign high attention scores to initial tokens with small semantics. This phenomenon extends beyond the scope of length generalization failure, indicating a more pervasive issue in Transformer models. We observe this "attention sink" behavior not only in auto-regressive language models but also in encoder Transformers such as BERT (see Section H), and Vision Transformers (ViTs) (Darcet et al., 2023), suggesting its broader prevalence in Transformer architectures. To mitigate the "attention sink" phenomenon, we propose the introduction of a learnable sink token during pre-training, and we support our findings with extensive ablation studies.

In parallel, Darcet et al. observed similar attention concentration on random background patch tokens in Vision Transformers, termed as "registers." These registers act as repositories for global image information. Their solution, adding dedicated "register" tokens, aims to balance attention distribution. Our finding of "attention sinks" parallels this concept. In our paper, the "attention sinks" are initial tokens that disproportionately attract attention from subsequent tokens. Introducing a dedicated sink token during pre-training prevents the model from inappropriately using content tokens as attention sinks, leading to more effective attention distribution. However, a key difference exists: "registers" in Vision Transformers function as global information holders within intermediate layers, whereas our "attention sinks" are positioned as initial tokens in autoregressive models. This positional variance suggests that the softmax function in attention computation might play a more fundamental role in the emergence of attention sinks.

## C   Accuracy on StreamEval with Increasing Query-Answer Line Distance

To assess StreamingLLM's handling of extended inputs, we evaluated the Llama-2-7B-32K-Instruct model on StreamEval, focusing on different query-answer line distances under various cache configurations. In StreamEval, each line consists of 23 tokens, making the line distances equivalent to token distances of $23 \times$ line distances. Accuracy was calculated by averaging results over 100 samples, with each sample comprising 100 queries. Table 7 illustrates that StreamingLLM retains accuracy when the token distance between the query and answer is within the cache size. However, accuracy diminishes as this distance increases and eventually drops to zero when it surpasses the cache capacity.

These results demonstrate that while StreamingLLM is effective in generating coherent text based on recent context, it cannot extend the context length of language models. These results also emphasize a broader challenge in current language models: their inability to fully utilize context information within the cache, a finding that aligns with the observations made by Liu et al..

Table 8: Performance comparison of StreamingLLM against the default truncation baseline in LongBench (Bai et al., 2023). The baseline truncates inputs to 1750 initial and 1750 final tokens. StreamingLLM 4+3496 uses 4 attention sink tokens and 3496 recent tokens, while StreamingLLM 1750+1750 uses 1750 tokens for both initial and recent segments.

| Llama2-7B-chat | Single-Document QA | | Multi-Document QA | | Summarization | |
|---|---|---|---|---|---|---|
| | NarrativeQA | Qasper | HotpotQA | 2WikiMQA | GovReport | MultiNews |
| Truncation 1750+1750 | 18.7 | 19.2 | 25.4 | 32.8 | 27.3 | 25.8 |
| StreamingLLM 4+3496 | 11.6 | 16.9 | 21.6 | 28.2 | 23.9 | 25.5 |
| StreamingLLM 1750+1750 | 18.2 | 19.7 | 24.9 | 32.0 | 26.3 | 25.9 |

## D  LONG-RANGE BENCHMARK EVALUATION

We evaluated StreamingLLM using the Llama-2-7B-chat model (max context length 4k) on Long-Bench (Bai et al., 2023), which encompasses three key NLP tasks: single-document QA (NarrativeQA (Kočiský et al., 2017) and Qasper (Dasigi et al., 2021)), multi-document QA (HotpotQA (Yang et al., 2018) and 2WikiMQA Ho et al. (2020)), and summarization (GovReport (Huang et al., 2021), MultiNews (Fabbri et al., 2019)). LongBench sets a default max sequence length of 3,500 tokens for the Llama-2-7B-chat model, truncating from the middle to preserve beginning and end information (1,750 tokens each). Table 8 shows that StreamingLLM with a 4+3496 cache configuration underperforms compared to the truncation baseline, likely due to the loss of crucial initial input prompt information. However, aligning the attention sink number to 1750 restores performance to the level of the text truncation baseline. These results corroborate the findings in Section C, demonstrating that StreamingLLM's effectiveness is contingent on the information within its cache, with in-cache performance comparable to the text truncation baseline.

## E  LLAMA-2-7B ATTENTION VISUALIZATION ON LONGER SEQUENCES

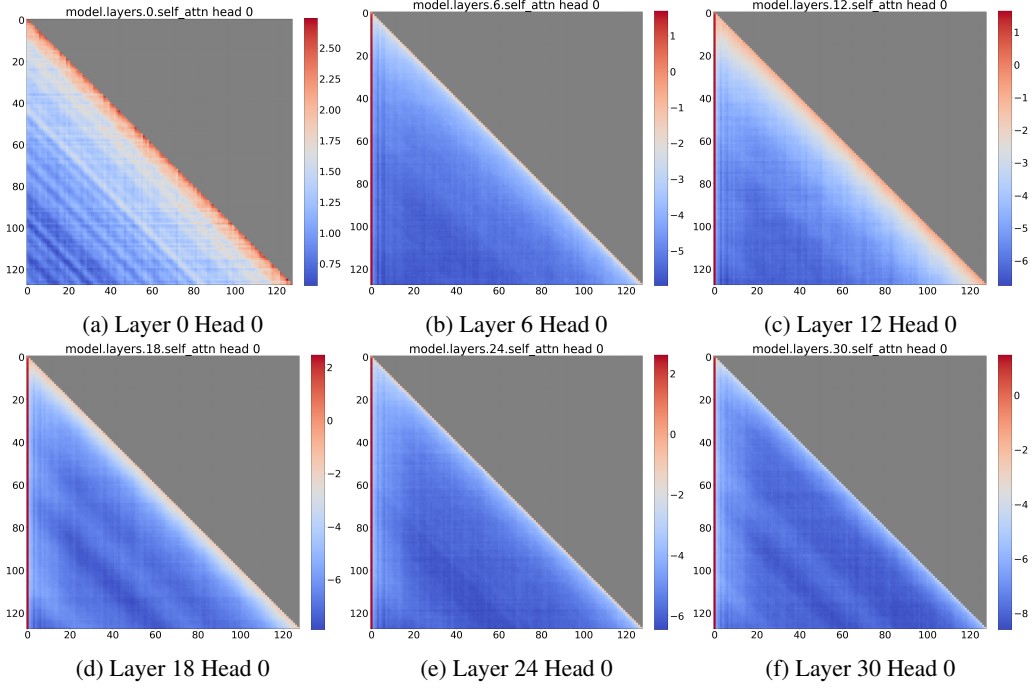

(a) Layer 0 Head 0  (b) Layer 6 Head 0  (c) Layer 12 Head 0

(d) Layer 18 Head 0  (e) Layer 24 Head 0  (f) Layer 30 Head 0

Figure 11: Visualization of the *average* attention logits in Llama-2-7B over 256 sentences, each with a length of 128.

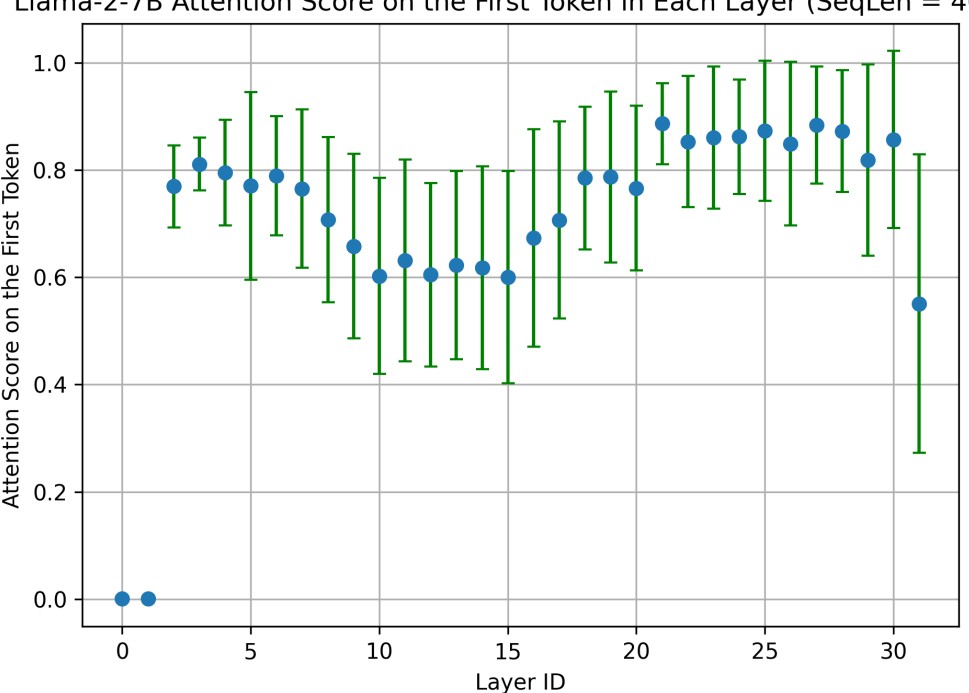

Figure 12: Visualization of attention scores (after SoftMax) on the first token across layers in Llama-2-7B. Attention Scores are the 4096th token's attention towards the first token in each layer. The error bars are the standard deviation of the first token's attention scores across different heads in one layer. Results are averaged over 256 sentences, each having a length of 4096 tokens.

Figure 2 visualizes the attention map of Llama-2-7B using short sequences (length of 16) for clarity. We further visualize the attention of Llama-2-7B on longer sequences (length of 128) in Figure 11. We find the observations on short sequences also hold on longer sequences, where the attention scores of the initial tokens are much higher than the rest of the tokens in most layers, regardless of the distance between the initial tokens and the tokens in the rest of the sequence. Because the longer the sequence, the thinner the attention sinks' scores are visualized on the heatmap. We further analyze the attention distribution on longer sequences (length of 4096) using a different method in Section F.

## F  QUATITATIVE ANALYSIS OF ATTENTION SINKS IN LONG INPUTS

Figures 2 and 13 illustrate the attention sink phenomenon using short sequences for clarity. Extending this analysis, Figure 12 demonstrates the distribution of attention scores (after SoftMax) towards the first token in lengthy inputs (sequence length of 4096). We average attention scores across 256 sequences, with each sequence comprising 4096 tokens. The plotted data represent the attention allocated by the 4096th token to the initial token in every layer. Notably, the attention scores for the first token are significantly high, often exceeding half of the total attention, except for the two bottom layers. This observation empirically substantiates the preferential focus on the first token by the majority of layers and heads, irrespective of other tokens' distances within the sequence. Such a trend underscores the critical role of the initial tokens in a sequence, as their removal has a huge impact on language model performance due to a large portion of the denominator in the SoftMax function being removed.

## G  LLAMA-2-70B ATTENTION VISUALIZATION

Figure 2 shows the attention visualization of Llama-2-7B, we further visualize the attention of Llama-2-70B in Figure 13. We find the observation on Llama-2-7B also holds on Llama-2-70B,

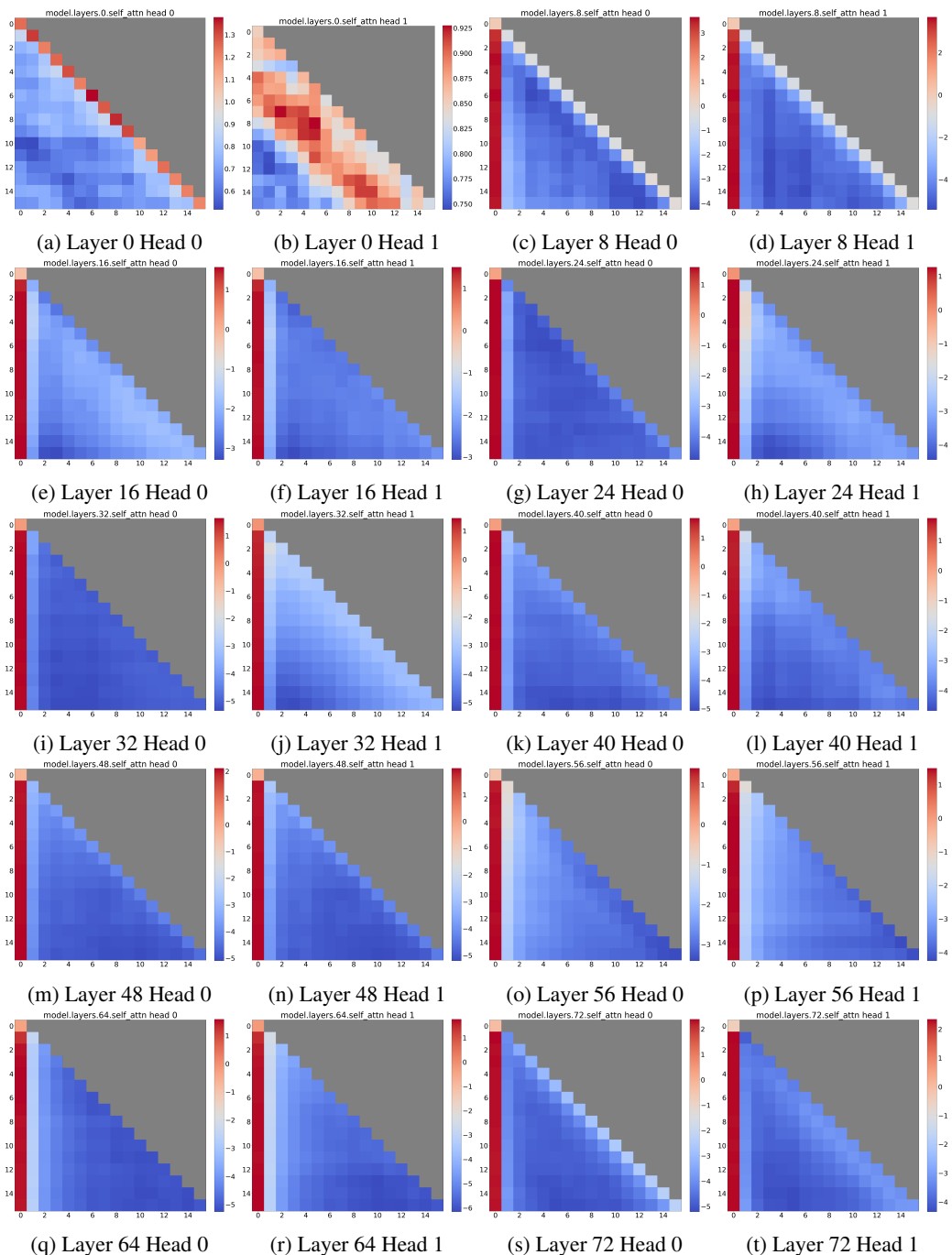

Figure 13: Visualization of the *average* attention logits in Llama-2-70B over 256 sentences, each with a length of 16.

where the attention scores of the initial tokens are much higher than the rest of the tokens in most layers.

## H ATTENTION SINKS IN ENCODER TRANSFORMERS

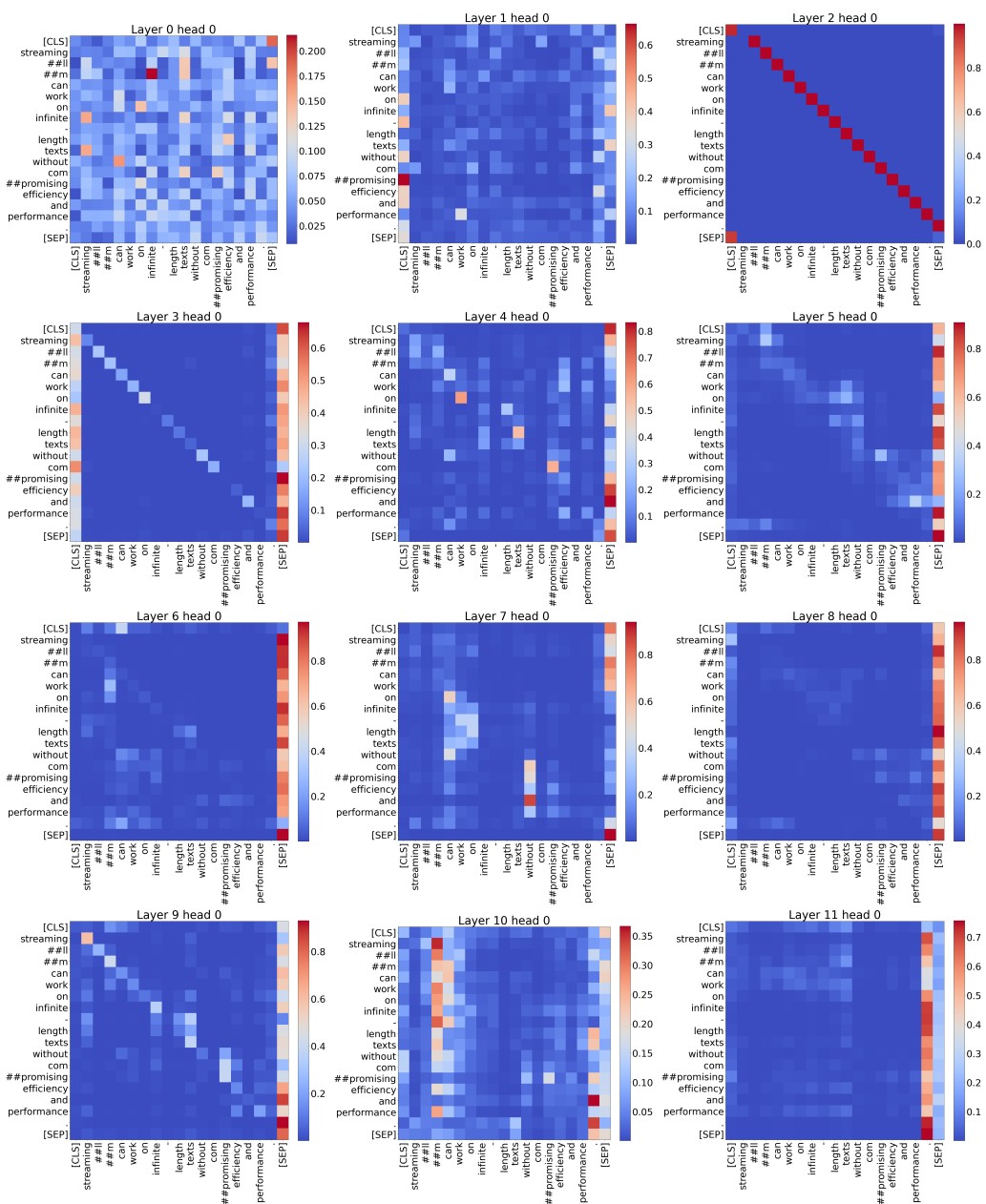

Figure 14: Visualization of attention maps for sentence *"StreamingLLM can work on infinite-length texts without compromising efficiency and performance."* in BERT-base-uncased.

In this paper, we mainly explore the attention sink phenomenon observed in autoregressive, decoder-only language models like GPT and Llama. Building upon the insights from Section 3.1, we propose that this phenomenon likely extends to other Transformer architectures, including encoder models such as BERT (Devlin et al., 2019) and ViT (Dosovitskiy et al., 2021). This assumption stems from the fact that these models share a similar Transformer structure and utilize SoftMax attention mechanisms. To substantiate our hypothesis, we analyze the attention patterns of BERT-base-uncased,

Table 10: Comparison of vanilla attention with prepending a zero token and a learnable sink token during pre-training. Cache config $x+y$ denotes adding $x$ initial tokens with $y$ recent tokens. Perplexity is evaluated on the first sample in the PG19 test set.

| Cache Config | 0+1024 | 1+1023 | 2+1022 | 4+1020 |
|---|---|---|---|---|
| Vanilla | 27.87 | 18.49 | 18.05 | 18.05 |
| + 1 Sink Token | 1235 | **18.01** | 18.01 | 18.02 |
| + 2 Sink Tokens | 1262 | 25.73 | 18.05 | 18.05 |

as depicted in Figure 14. Our findings reveal that BERT-base-uncased exhibits the attention sink phenomenon, characterized by disproportionately high attention scores assigned to the `[SEP]` token in most layers. This indicates that the model consistently relies on the omnipresent `[SEP]` token as a focal point for attention. Furthermore, concurrent research by Darcet et al. identifies similar attention spikes in Vision Transformers, attributed to random background patch tokens acting as "registers" for global image information. We contend that these "registers" are analogous to the attention sink phenomenon we observed, suggesting that this is a universal characteristic across all Transformer models.

# I USING MORE SINK TOKENS IN THE PRE-TRAINING STAGE

Section 3.3 illustrated that incorporating a single dedicated sink token in the pre-training stage doesn't affect model performance but enhances streaming performance by centralizing attention sinks to one token. This section delves into whether adding additional sink tokens during pre-training could further optimize the performance of pre-trained language models.

As depicted in Figure 15, our experiments show that incorporating either one or two sink tokens during pre-training results in pre-training loss curves that closely resemble those of the baseline (vanilla) model. However, as detailed in Table 9, the introduction of a second sink token does not yield substantial improvements in performance across most benchmark tasks.

Further analysis, as shown in Table 10, reveals that the inclusion of additional sink tokens does not enhance streaming performance. Interestingly, the model appears to rely on both sink tokens to maintain stable streaming performance. These findings suggest that while a single sink token is adequate for improving streaming performance, adding more sink tokens does not lead to further enhancements in overall language model performance. This contrasts with findings in Vision Transformers (ViT) (Darcet et al., 2023), where multiple "registers" have been found to be beneficial.

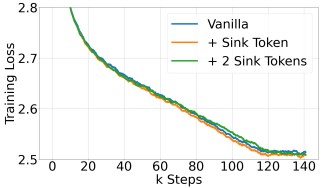

Figure 15: Pre-training loss curves of models with 0, 1, and 2 sink tokens.

Table 9: Zero-shot accuracy (in %) across 7 NLP benchmarks, including ARC-[Challenge, Easy], HellaSwag, LAMBADA, OpenbookQA, PIQA, and Winogrande.

| Methods | ARC-c | ARC-e | HS | LBD | OBQA | PIQA | WG |
|---|---|---|---|---|---|---|---|
| Vanilla | 18.6 | 45.2 | 29.4 | 39.6 | 16.0 | 62.2 | 50.1 |
| + 1 Sink Token | **19.6** | **45.6** | **29.8** | **39.9** | **16.6** | 62.6 | **50.8** |
| + 2 Sink Tokens | 18.7 | 45.6 | 29.6 | 37.5 | 15.8 | **64.3** | 50.4 |

