# OpenReview forum: "Efficient Streaming Language Models with Attention Sinks"
_ICLR.cc/2024/Conference — ICLR 2024 poster_

### Official Review · Reviewer_MUgn · 2023-10-31

**Soundness:** 4 excellent
**Presentation:** 3 good
**Contribution:** 4 excellent
**Rating:** 8
**Confidence:** 4

**Summary:**

The paper introduces StreamingLLM, a framework that enables large language models (LLMs) to handle infinite sequence lengths without fine-tuning. The paper makes the following contributions:

- It observes the attention sink phenomenon, which means that LLMs tend to allocate a lot of attention to the initial tokens, regardless of their semantic importance.
- It proposes the StreamingLLM framework, which leverages the attention sink phenomenon to retain only the most recent tokens and a few attention sinks, discarding intermediate tokens. This allows LLMs to generate coherent text from recent tokens without a cache reset.
- It suggests adding a placeholder token as a dedicated attention sink during pre-training, which can further improve the streaming performance of LLMs.
- It demonstrates that StreamingLLM can enable models like Llama-2, MPT, Falcon, and Pythia to perform efficient language modeling with up to 4 million tokens.

**Strengths:**

- This paper studies an important topic/setting on the long context of large language models, how to enhance LLMs to handle long/infinite sequence without fine-tuning.
- Attention sink phenomenon is remarkably interesting. It was found that retaining the KV of initial tokens can significantly improve the performance of window attention.
- This paper poses comprehensive experimental investigations on different large language models, and it also shows that StreamingLLM outperforms other baselines by up to 22.2× speedup in streaming settings.
- The paper is well written, clear to follow.

**Weaknesses:**

See Questions Section

**Questions:**

- The authors focus on LLMs that are autoregressive, meaning that they generate text one token at a time, conditioned on the previous tokens. Dose the attention sink phenomenon exist in other types of models, such as bidirectional models (e.g., BERT) or non-transformers models?
- Can attention sink be applied to the pre-training stage, retaining more placeholders from scratch to further improve the performance?
- Does this phenomenon also exist in larger models like Falcon-40B, LLaMa2-70B?
- Does the Group Query Attention structure have any impact?
- If you combine proposed Attention Sink with Sliding Window w re-computation, I wonder whether it performs better than re-computation.

---

> ### Author Response · Authors · 2023-11-21
> **Response to Reviewer MUgn**
>
> **Q: Attention Sink Phenomenon in Non-Autoregressive Models**
>
> A: The attention sink phenomenon is not exclusive to autoregressive models, which we focused on in our study. Appendix Section H of our revised paper shows that encoder models, like BERT, also exhibit this phenomenon. Concurrent to our work, Darcet et al. observe similar attention concentration on random background patch tokens in Vision Transformers, termed "registers." This suggests that the tendency to allocate significant attention to sink tokens might be a broader characteristic of transformer architectures, irrespective of whether they are autoregressive or bidirectional.
>
> ---
>
> **Q: Using more Attention Sinks in the Pre-Training Stage**
>
> A: Thanks for your suggestion. We added an experiment where we pre-trained the 160M language model with two sink tokens, detailed in Appendix Section I. Our experiments indicated that adding more than one attention sink token does not significantly enhance performance. This finding suggests that a single dedicated attention sink during pre-training is sufficient to leverage the benefits of this phenomenon.
>
> ---
>
> **Q: Attention Sink Phenomenon in Larger Models**
>
> A: Yes, this phenomenon persists in larger models like Falcon-40B and Llama2-70B. We conducted perplexity experiments in Figure 5 to confirm this and included visualizations in Appendix Section G of our manuscript. These results demonstrate that the attention sink phenomenon exists across different model sizes, and StreamingLLM is scalable to all of them.
>
> ---
>
> **Q: Impact of Multi/Group Query Attention Structure**
>
> A: The attention sink phenomenon was still observable in our experiments, including models like Falcon-40B using multi-query attention and Llama-2-70B using group-query attention. This suggests that different attention structures, such as group query attention, do not negate the presence of attention sinks (Appendix G). StreamingLLM effectively addresses this issue despite the architecture differences (Figure 5), demonstrating its versatility across various model architectures.
>
> ---
>
> **Q: Combining Attention Sink with Sliding Window with Re-computation**
>
> A: Combining attention sinks with sliding windows with re-computation does not provide additional benefits over re-computation alone. This is because the sliding window with re-computation inherently includes full attention on truncated text, meaning attention sinks are always present. This is why sliding windows with re-computation perform better than the window attention baseline. Our approach with StreamingLLM focuses on efficiency and speed, significantly reducing the computational complexity compared to this baseline.
>
> [1] Darcet, T., Oquab, M., Mairal, J., & Bojanowski, P. (2023). Vision Transformers Need Registers. *ArXiv*. /abs/2309.16588

---

> > ### Author Response · Authors · 2023-11-22
> > **Follow up**
> >
> > Thanks again for reviewing our work! Please let us know if we have adequately addressed your questions and concerns.
> >
> > Authors

---

> > > ### Author Response · Authors · 2023-11-22
> > > **Kind Request for Prompt Feedback as Discussion Period Concludes**
> > >
> > > We are deeply grateful for your valuable insights on our paper. With the discussion period closing in less than a day, we would greatly appreciate your thoughts on our recent response. Your feedback is vital to ensure we have addressed your concerns adequately.
> > >
> > > Thank you for your time and consideration!
> > >
> > > Authors

---

### Official Review · Reviewer_STcB · 2023-11-01

**Soundness:** 3 good
**Presentation:** 4 excellent
**Contribution:** 3 good
**Rating:** 6
**Confidence:** 4

**Summary:**

This paper demonstrates that the first few token(s), called attention sink, take most of the attention scores in large language models and propose StreamingLLM by keeping them in the KV cache. StreamingLLM achieves significant speedup compared to the baseline of sliding window attention with recomputation while having reasonable language modeling perplexity and zero-shot accuracy on several NLP benchmarks. Additionally, the authors pre-trained a small language model with a single learnable sink token so that StreamingLLM can only use one token for attention sink.

**Strengths:**

Improving the efficiency of language modeling with long text input is a practically important and very timely topic. The paper is well motivated and structured in that it first introduces problems, suggests an easy and effective solution, and validates it with experiments and required ablation studies.

**Weaknesses:**

StreamingLLM is compared with only very basic baselines (i.e., (a) dense attention, (b) window attention, and (c) sliding window with re-computation) without alternative ways to use LLMs with longer inputs. The authors argue StreamingLLM’s 22.2x speedup compared to (c), but it looks too extreme because there could be a sweet spot that could achieve reasonable ppl and inference speed by sliding window with appropriate strides (less frequent re-computation instead of every step).

The paper argues that StreamingLLM can handle infinite sequence length, but again the abovementioned basic baselines on PG-19 are used to show this. Experiments on benchmark datasets that truly require long context modeling are necessary. Also, providing numbers of SoTA models on PG-19 could be helpful in understanding how absolutely StreamingLLM performs well, though accuracy might not be the main interest of this paper. Particularly, using a small size cache (not beyond the training sequence length) makes it suspicious whether StreamingLLM is capturing long context information well. I guess it will have difficulty utilizing information from tokens that do not belong to the cache at the prediction step, and this can be easily checked with synthetic datasets (e.g., Little Retrieval Test).

There have been several previous works that observe position bias on attention patterns in transformers. Also, there are existing architectures, including LongFormer, BigBird, and ETC, that have global tokens for attention, though most of them are encoder-based language models. It is worth having a discussion on them.

**Questions:**

Could you elaborate more on why (1) window attention blows up and (2) attention sink exists?
I think current explanations are mostly empirical and not theoretically grounded. Specifically, I agree that attention scores could be skewed toward earlier tokens due to the autoregressive nature of language models. However, why very few (less than 4) tokens are enough to cover most attention scores?

Visualization on a short sequence (length 16) in Figure 2 might not be enough. Does the same phenomenon occur for longer sequences, which is our main interest? I guess some important tokens with high attention scores might be shown up for each text (of course, it will not be shown if you average per position over different texts). Do you have any quantitative values on how much attention sink occupies the attention scores (especially for longer inputs)?

In Figure 3, MPT has different patterns from other LLMs. Is it because MPT uses ALiBi and others use RoPE? Any explanation?

Regarding pre-training with a sink token, I think reducing 4 -> 1 is not important considering the large size of the cache. Am I missing? It could be marginal but why is a model with a sink token consistently better than the vanilla one? An LM with 160M parameters is used, and I understand this is because of the academic budget. I am just wondering if the same conclusion can be guaranteed for larger LMs.

In Figure 10,  why do two options have different memory usage for 4096 cache size?

This paper uses the attention sink for efficient streaming of LLMs. However, my one aspect is having attention sink is a kind of artifact of autoregressive LLMs not fully utilizing its representation power. Can we make/train LLMs not to have the attention sink for better performance?

There is almost similar concurrent work: LM-Infinite (https://arxiv.org/abs/2308.16137). It is not required, but how do you compare your work with this?

---

> ### Author Response · Authors · 2023-11-21
> **Response to Reviewer STcB (1/2)**
>
> **Q: Comparison with Sliding Window with Re-computation**
>
> A: We appreciate your point about comparing StreamingLLM with sliding window approaches with varying strides. Our 22.2x speedup comparison was established under identical conditions to ensure a fair evaluation. StreamingLLM's linear $O(N)$ complexity contrasts sharply with the sliding window's quadratic $O(N^2)$ complexity. While adjusting stride lengths in sliding windows may offer some efficiency, it introduces context size and computational workload variability, resulting in inconsistent decoding times and greater computational demands during re-computation phases. This inconsistency is a significant drawback compared to StreamingLLM's consistent performance and computational efficiency.
>
> ---
>
> **Q: Misunderstanding of Long-Context**
>
> A: It is crucial to highlight that StreamingLLM is not designed to extend the context window or enhance long-term memory capabilities. Instead, its primary aim is efficient text generation from recent tokens without frequent cache refreshes. We have included further clarification on this in Appendix Section A, to articulate StreamingLLM's intended application better and differentiate it from long-context models.
>
> ---
>
> **Q: Missing Discussion with Related Works**
>
> A: Thank you for pointing out the need for a more detailed comparison with related works. We have added Appendix Section B to include a comprehensive discussion on architectures like LongFormer, BigBird, and ETC that employ global tokens for attention. This enhanced comparison delineates the unique aspects of our approach, emphasizing its simplicity and compatibility with existing transformer architectures while also recognizing the foundational concepts established by these earlier models.
>
> ---
>
> **Q: Why Does Window Attention Blow Up?**
>
> A: The issue of window attention blowing up is linked to the dynamics of the SoftMax function used in attention mechanisms. As detailed in Appendix Section F, more than half of attention scores in most layers and heads are allocated to the first token. Over half of the SoftMax function's denominator is removed when these initial tokens are evicted. This significant alteration in the attention score distribution deviates from the model's pre-training conditions, leading to a drastic change in the SoftMax distribution. Consequently, this results in model performance degradation, as the attention mechanism no longer functions as it was trained to.
>
> ---
>
> **Q: Why Does Attention Sink Exist? Why Are Few Tokens Enough?**
>
> A: The attention sink phenomenon primarily arises from the characteristics of the SoftMax function used in attention mechanisms. SoftMax necessitates that all attended tokens’ attention scores add up to one, forcing the model to distribute some attention across contextual tokens even when the current token embedding already contains ample self-relevant information for prediction or lacks a strong correlation with many preceding keys. As a result, the model learns to allocate excessive attention scores to certain tokens.
>
> Given the sequential nature of autoregressive language modeling, initial tokens are consistently visible to all subsequent tokens. In contrast, later tokens have visibility restricted to fewer subsequent tokens. This visibility pattern predisposes initial tokens to become attention sinks as they are capable of absorbing unnecessary attention from a wider range of tokens.
>
> We empirically observe that the number of tokens required to function as effective attention sinks are relatively small. A few tokens are typically sufficient to absorb excess attention. The precise reason why this limited number of tokens is adequate remains an area for future theoretical research.
>
> ---
>
> **Q: Attention Map Visualization on Long Sequences**
>
> A: The attention sink phenomenon persists across all sequence lengths. For clarity in visualization, we focused on shorter sequences in the paper’s figure. In Appendix Section E of our updated manuscript, we include attention map visualizations for longer sequences, illustrating this phenomenon's consistency. Appendix Section F also includes quantitative results on the ratio of attention sinks’ attention scores on long sequences with 4096 tokens.

---

> ### Author Response · Authors · 2023-11-21
> **Response to Reviewer STcB (2/2)**
>
> **Q: Quantitative Results on Attention Sinks’ Attention Scores**
>
> A: Thank you for your suggestion! We conducted experiments to quantify the attention sinks' score ratio in the Llama models in Appendix Section F. We plotted attention allocated by the 4096th token to the initial token in every layer. Notably, the attention scores for the first token are significantly high, often exceeding half of the total attention, except for the two bottom layers. This observation empirically substantiates the preferential focus on the first token by most layers and heads, irrespective of other tokens' distances within the sequence. Such a trend underscores the critical role of the initial tokens in a sequence, as their removal greatly impacts language model performance due to a large portion of the denominator in the SoftMax function being removed.
>
> ---
>
> **Q: Explanation for MPT’s Different Perplexity Curve Pattern**
>
> A: The general behavior of the MPT model aligns with that observed in RoPE models, particularly in how perplexity sharply rises when the input length surpasses the pre-training length in dense attention scenarios and similarly when it exceeds the cache size in window attention contexts.
>
> However, the specific perplexity curve pattern in MPT differs due to a combination of factors, including variations in pre-training approaches, model architecture, training datasets, positional encoding, and activation functions. A comprehensive analysis of these discrepancies and their underlying causes is complex and falls outside the scope of our current work. We acknowledge this as an area for future research to explore in more detail.
>
> ---
>
> **Q: Purpose of Pre-training Experiments**
>
> A: Our pre-training experiments aimed to test whether introducing a globally visible sink token would direct excessive attention to this token, thus eliminating other attention sinks. This experiment validated our hypothesis, showing that a single token is enough to recover the streaming perplexity, and the model no longer relies on other initial tokens. While our study used 160M LM due to budget constraints, we expect our pre-training solution to generalize to larger models. Future research with more resources can test our findings in larger LLMs.
>
> ---
>
> **Q: Different Memory Usage in StreamingLLM vs. Sliding Window with Re-computation**
>
> A: The slightly higher memory usage in sliding windows with recomputation is attributed to the need for intermediate variables for the full QxK quadratic attention computation. In contrast, StreamingLLM efficiently computes only the qxK attention, reducing overall memory requirements.
>
> ---
>
> **Q: Possibility of LLMs Without Attention Sink**
>
> A: Your suggestion of potentially training LLMs without attention sinks to enhance performance is intriguing and warrants further exploration. The involvement of the SoftMax function in the formation of attention sinks is a critical aspect to consider. Investigating alternatives to SoftMax for attention computation could indeed be a promising avenue. Additionally, incorporating a trainable denominator within the SoftMax function presents another viable approach.
>
> ---
>
> **Q: Comparison with Concurrent Work (LM-Infinite)**
>
> A: LM-Infinite (Han et al., 2023) offers valuable theoretical insights into LLM length generation failure, identifying three out-of-distribution factors. Inspired by this analysis, their approach involves employing a ""$\Lambda$-shaped" attention pattern and reconfiguring position encoding distances to enhance length generalization in LLMs. This approach bears a resemblance to our methodology.
>
> Distinctly, our work uncovers the "attention sink" phenomenon by looking into the attention maps of LLMs, wherein Transformer models assign high attention scores to initial tokens with minor semantics. This phenomenon extends beyond the scope of length generalization failure, indicating a more pervasive issue in Transformer models. We observe this "attention sink" behavior not only in auto-regressive language models but also in encoder Transformers such as BERT (see Appendix Section H) and Vision Transformers (Darcet et al., 2023), suggesting its broader prevalence in Transformer architectures. To mitigate the “attention sink” phenomenon, we propose the introduction of a learnable sink token during pre-training, and we support our findings with extensive ablation studies. We believe our work and LM-Infinite contribute unique insights to the same topic, providing valuable empirical evidence and theoretical perspectives that can inform future developments in Transformer model architecture.
>
> We have added this discussion in Appendix Section B in our revised paper.
>
> [1] Darcet, T., Oquab, M., Mairal, J., & Bojanowski, P. (2023). Vision Transformers Need Registers. *ArXiv*. /abs/2309.16588
>
> [2] Han, C., Wang, Q., Xiong, W., Chen, Y., Ji, H., & Wang, S. (2023). LM-Infinite: Simple On-the-Fly Length Generalization for Large Language Models. *ArXiv*. /abs/2308.16137

---

> > ### Author Response · Authors · 2023-11-22
> > **Follow up**
> >
> > Thanks again for reviewing our work! Please let us know if we have adequately addressed your questions and concerns.
> >
> > Authors

---

> > > ### Author Response · Authors · 2023-11-22
> > > **Kind Request for Prompt Feedback as Discussion Period Conclude**
> > >
> > > We are deeply grateful for your valuable insights on our paper. With the discussion period closing in less than a day, we would greatly appreciate your thoughts on our recent response. Your feedback is vital to ensure we have addressed your concerns adequately.
> > >
> > > Thank you for your time and consideration!
> > >
> > > Authors

---

### Official Review · Reviewer_obg3 · 2023-11-01

**Soundness:** 3 good
**Presentation:** 4 excellent
**Contribution:** 3 good
**Rating:** 8
**Confidence:** 4

**Summary:**

The paper focuses on the challenge of efficiently scaling and generalizing beyond the training sequence length for large language models. Specifically, they propose to cache and attend to the initial tokens of the sequence in addition to the most recent tokens used in window attention. The main motivation behind this proposal is the "attention sink" phenomenon that is caused by the strong attention scores toward initial tokens that are accumulated as the sequence increases. Since this method introduces this simple change of attending to the initial tokens, it can be applied to any LLM; they also show it is possible to train the model with a single token as attention sink for further efficiency. Experiments on a models of vaying model size show that it allows them to generalize far beyond their context window and it is much faster than sliding window with recomputation.

**Strengths:**

- Several methods have been introduced in the past for generalizing to longer context lengths but most of them require training the model from scratch or through continued training. The observation that this paper is making about the attention sink phenomenon is fascinating and inspires a simple solution that requires no additional training that prior work has overlooked.
- The paper is positioned well wrt. prior length generalization work and shows how different relative positional embedding methods such as RoPE anD Alibi can be combined with attention sinks. The fact that it can be combined with any of these methods with minimal effort makes it potentially very impactful; virtually any transformer model can be modified to handle large context sizes in this way.
 - The exposition is clear and experimental part well executed. The empirical analysis based on actual attention logits and the theoretical explanation of why it happens is convincing and intuitive.
- Evaluation covers models of different sizes and different metrics (perplexity and downstream performance) and shows that proposed attention is effective across all sequence lengths.

**Weaknesses:**

- The proposed attention is reminiscent of methods that introduce sparse attention patterns such as Sparse Transformer but there is no in-depth discussion that draws a connection.
- Even though attention sinks maintains the perplexity levels in check for extremely large sequence lengths, the paper does not study in detail on how good utilization of the context the model is doing.

**Questions:**

- Have you considered evaluating on a benchmark for long-range tasks such as LongBenh, MuLD or Long-range arena? It would help quantify limitations of the proposed attention in terms of context utilization.

---

> ### Author Response · Authors · 2023-11-21
> **Response to Reviewer obg3**
>
> **Q: Missing Discussion with Related Works**
>
> A: We value your feedback on contextualizing our work with prior literature. In response, we have added Appendix Section B to include a comprehensive discussion that contrasts our attention sink mechanism with sparse attention patterns found in models like Sparse Transformer, LongFormer, and BigBird. This enhanced comparison delineates the unique aspects of our approach, emphasizing its simplicity and compatibility with existing transformer architectures while also recognizing the foundational concepts established by these earlier models.
>
> ---
>
> **Q: Utilization of the Context**
>
> A: We acknowledge the importance of evaluating how effectively the model utilizes context, especially when introducing attention sinks. In our initial experiments, we observed satisfactory context utilization within the cache limits, shown as the language modeling performance is comparable with the oracle baseline, i.e., sliding window with re-computation.
>
> To analyze the extent of context utilization beyond these cache limits, we conducted further experiments, detailed in Appendix Section C, to assess context utilization when dealing with cache-exceeding information. The findings corroborate that StreamingLLM’s capacity to utilize context is limited to the current cache, with no ability to leverage evicted tokens. It's crucial to emphasize that StreamingLLM's primary aim is not to augment the context window or boost long-term memory retention. Instead, its strength lies in generating coherent text from recent tokens without necessitating cache refreshes.
>
> Moreover, our experiments demonstrate that StreamingLLM can be integrated with recent advancements in context extension methods. With context-extended models, we can use larger caches to maintain a broader range of recent tokens in StreamingLLM. For a demonstration of StreamingLLM's integration with models like LongChat-7B-v1.5-32K and Llama-2-7B-32K-Instruct, please refer to Figure 9 in our paper.
>
> ---
>
> **Q: Long-Range Benchmark Evaluation**
>
> A: Following your recommendation, we have evaluated StreamingLLM on long-range tasks using LongBench (Bai et al., 2023), which includes diverse NLP challenges. Our focus was on assessing how StreamingLLM, particularly using the Llama-2-7B-chat model with a maximum context length of 4k, performs on tasks like single-document QA (NarrativeQA and Qasper), multi-document QA (HotpotQA and 2WikiMQA), and summarization (GovReport and MultiNews).
>
> The results in Appendix Section D of our revised manuscript show that StreamingLLM's performance is on par with the text truncation baseline. However, it's crucial to note that StreamingLLM is not primarily designed to enhance the utilization of long texts. Its strength lies in efficiently utilizing recent information without the need for cache refreshes or extensive memory. Therefore, while we acknowledge this limitation, addressing it falls outside the scope of our current work and is a potential area for future research.
>
> [1] Bai, Y., Lv, X., Zhang, J., Lyu, H., Tang, J., Huang, Z., Du, Z., Liu, X., Zeng, A., Hou, L., Dong, Y., Tang, J., & Li, J. (2023). LongBench: A Bilingual, Multitask Benchmark for Long Context Understanding. *ArXiv*. /abs/2308.14508

---

> > ### Author Response · Authors · 2023-11-22
> > **Follow up**
> >
> > Thanks again for reviewing our work! Please let us know if we have adequately addressed your questions and concerns.
> >
> > Authors

---

> > > ### Author Response · Authors · 2023-11-22
> > > **Kind Request for Prompt Feedback as Discussion Period Concludes**
> > >
> > > We are deeply grateful for your valuable insights on our paper. With the discussion period closing in less than a day, we would greatly appreciate your thoughts on our recent response. Your feedback is vital to ensure we have addressed your concerns adequately.
> > >
> > > Thank you for your time and consideration!
> > >
> > > Authors

---

> > ### Comment · Reviewer_obg3 · 2023-11-23
> > **Response to rebuttal**
> >
> > Thank you for the response. I appreciate the additional experiments and for making the point clear about this limitation of attention sinks; would be great if this is reflected in the main text as well.

---

> > > ### Author Response · Authors · 2023-11-23
> > > **Thanks for the confirmation**
> > >
> > > Thank you for your confirmation! In the next revision, we will clarify the limitation in the main text.
> > >
> > > Authors

---

### Official Review · Reviewer_PzfP · 2023-11-06

**Soundness:** 4 excellent
**Presentation:** 4 excellent
**Contribution:** 3 good
**Rating:** 8
**Confidence:** 5

**Summary:**

This work proposes the addition of a placeholder token, called a token sink, to language models.This is motivated by the observation that significant attention mass is placed at the initial token positions in a sequence and performance deteriotes whenever tokens in this position are dropped/evicted. The authors carry out extensive experiments to validate this claim and make a convincing argument for the placement of a special token in the initial position, whose purpose is to help regulate and channel the attention mass across layers. This leads to stable language model performance at million-scale sequence lengths.

**Strengths:**

The paper is well written, the problem is well motivated with many experiments used to help make a convincing claim.

All code and datasets are made available, allowing for easy reproducibility and validation of claims made in this work.

**Weaknesses:**

The claims on long horizon performance need additional clarity and evaluation. I am not sure how practically useful this approach is for long-horizon language modelling.. For example, at million-scale sequence lengths, when intermediate tokens are evicted, how does performance translate relative to the evicted tokens? Doesnt it mean, the model is only practically useful for information covered at the beginning of the sequence and at the end?

There is a missing section on the broader societal impacts of this work.

**Questions:**

It is interesting to see how concentrated attention gets on the initial tokens as you go deeper into the model, could it be said that these tokens are carrying out some form of global memory/attention operation?

How does this work relate to [Vision Transformers Need Register](https://arxiv.org/abs/2309.16588v1).. This statement from this work reminds me of the registers paper; "As a result,
initial tokens are more easily trained to serve as attention sinks, capturing unnecessary attention"

Looking at the StreamEval benchmark, it would also be good to have an idea of how performance decays with length. How does it perform when you ask it a question related to line 1k, after it has seen 10/100k lines. This gives a better idea of long-context/sequence performance profile than consistently quering only 20 lines prior. I would be willing to improve my score if i had better insight of how this method performed on such a benchmark over (extremely) long horizons. How it performs relative to the evicted tokens.


Update: Authors addressed all my questions.

---

> ### Author Response · Authors · 2023-11-21
> **Response to Reviewer PzfP**
>
> **Q: Practical Application of StreamingLLM**
>
> A: Thank you for your insightful question! StreamingLLM is especially suited for applications requiring continuous, compute- and memory-efficient operation, such as a conversational AI assistant. Unlike previous methods that lose context or need time-intensive recomputation of KV states, StreamingLLM maintains conversational continuity and context without cache refresh. To illustrate, we emulate multi-round dialogues in real-world settings using instruction-tuned LLMs (Section 4.3). Our results demonstrate StreamingLLM's ability to produce quality outputs without KV recomputation or cache refresh. We have included a discussion in Appendix Section A in our revised paper.
>
> ---
>
> **Q: Discussion on Broader Societal Impacts**
>
> A: We acknowledge the importance of discussing the broader societal impacts of our research. To address this, we have expanded our manuscript to include thoroughly examining ethical considerations, potential societal benefits, and risks associated with StreamingLLM. This comprehensive discussion can be found in Appendix Section A of the revised paper.
>
> ---
>
>
> **Q: Interpreting Attention Sinks as Global Memory**
>
> A: We value your perspective on attention sinks. However, we think interpreting attention sinks in autoregressive language models as global memory may not be accurate. In these models, tokens are limited to attending to preceding tokens, not subsequent ones. Thus, the attention sinks, being initial tokens, don't aggregate information from future tokens and can't serve as global memory.
>
> ---
>
>
> **Q: Relation to “ViTs Need Registers” (Darcet et al., 2023)**
>
> A: Your connection to the “ViTs Need Registers” paper is insightful. Their study observed attention spikes on random background patch tokens in Vision Transformers, identified as "registers" holding global image information. They proposed adding dedicated "register" tokens to provide storage space and mitigate these attention spikes. Similarly, in our work, the "attention sinks" are initial tokens that attract disproportionate attention scores from subsequent tokens. We discovered that introducing a dedicated sink token during pre-training can prevent the model from using actual content tokens as attention sinks, resulting in cleaner attention maps.
>
> Our analysis suggests that the softmax function in attention mechanisms plays a crucial role in the emergence of both phenomena, as it causes models to concentrate excess attention on specific tokens. However, there's a notable distinction: while "registers" in Vision Transformers serve as global information repositories within intermediate layers, our "attention sinks" are initial tokens in autoregressive models. This positional difference means they don't provide similar global information storage, indicating that the softmax function's role in attention computation may be a more fundamental factor in developing attention sinks.
>
> This comparison underscores the similarities and differences in how attention is managed across different transformer models and provides a deeper understanding of how attention mechanisms can be optimized in future models. We have included this discussion in Appendix Section B.
>
> ---
>
> **Q: Accuracy on StreamEval with Increasing Line Distance**
>
> A: In response to your feedback, we conducted further experiments to evaluate StreamingLLM's performance with extended dependencies. As detailed in Appendix Section C of our revised manuscript, StreamingLLM maintains accuracy when the token distance between the query and answer is within the cache size. However, accuracy diminishes as this distance increases and eventually drops to zero when it surpasses the cache capacity.
>
> These results demonstrate that while StreamingLLM effectively generates coherent text based on recent context, it cannot extend the context length of language models. These results also emphasize a broader challenge in current language models: their inability to utilize context information within the cache fully, a finding that aligns with the observations made by the “Lost-in-the-Middle” paper (Liu et al., 2023).
>
> Including these results in our manuscript provides a clearer picture of both the strengths and limitations of StreamingLLM, particularly in scenarios requiring the management of extremely long contexts.
>
> [1] Liu, N. F., Lin, K., Hewitt, J., Paranjape, A., Bevilacqua, M., Petroni, F., & Liang, P. (2023). Lost in the Middle: How Language Models Use Long Contexts. ArXiv. /abs/2307.03172
>
> [2] Darcet, T., Oquab, M., Mairal, J., & Bojanowski, P. (2023). Vision Transformers Need Registers. ArXiv. /abs/2309.16588

---

> > ### Author Response · Authors · 2023-11-22
> > **Follow up**
> >
> > Thanks again for reviewing our work! Please let us know if we have adequately addressed your questions and concerns.
> >
> > Authors

---

> > > ### Author Response · Authors · 2023-11-22
> > > **Kind Request for Prompt Feedback as Discussion Period Concludes**
> > >
> > > We are deeply grateful for your valuable insights on our paper. With the discussion period closing in less than a day, we would greatly appreciate your thoughts on our recent response. Your feedback is vital to ensure we have addressed your concerns adequately.
> > >
> > > Thank you for your time and consideration!
> > >
> > > Authors

---

### Author Response · Authors · 2023-11-21
**Revision Summary**

We sincerely thank all reviewers and ACs for their valuable time and insightful reviews of our paper. We are pleased that our contributions have been recognized and appreciate the constructive feedback provided. In response, we have made significant revisions to our manuscript, enriching it with additional content and detailed discussions. We added **a new appendix** to enhance the comprehensiveness of our work on StreamingLLM. Key additions include:

- **Practical Application of StreamingLLM (Appendix Section A)**: Discussed the model's practical applications in scenarios like conversational AI, emphasizing its efficiency and recent context retention **(Reviewers PzfP and STcB)**.

- **Broad Societal Impacts (Appendix Section A)**: Discussed on the societal implications and ethical considerations of our research **(Reviewer PzfP)**.

- **Additional Related Works and Comparison with Concurrent Works (Appendix Section B)**: Included a detailed comparison with related works and concurrent studies, such as LM-Infinite and ViTs Need Registers, to contextualize our approach within the broader research landscape **(Reviewers PzfP, obg3, and STcB)**.

- **Experiment on StreamEval with Increasing Line Distance (Appendix Section C)**: Conducted additional experiments to assess performance over extended dependencies, focusing on context utilization and long-context handling **(Reviewers PzfP and obg3)**.

- **Experiment on LongBench (Appendix Section D)**: Evaluated StreamingLLM in long-range tasks using LongBench **(Reviewer obg3)**.

- **Visualization of Attention Maps on Longer Sequences (Appendix Section E)**: Provided visualizations for longer sequences to demonstrate the attention sink phenomenon **(Reviewer STcB)**.

- **Quantitative Analysis of Attention Sinks’ Attention Scores (Appendix Section F)**: Presented quantitative results on attention sinks' score ratios, particularly for longer inputs **(Reviewer STcB)**.

- **Visualization of Attention Maps in Llama2-70B Model (Appendix Section G)**: Included visualizations showing attention patterns in larger models like Llama2-70B **(Reviewer MUgn)**.

- **Visualization of Attention Maps in BERT Model (Appendix Section H)**: Showcased attention maps in non-autoregressive models like BERT, highlighting the ubiquity of the attention sink phenomenon **(Reviewer MUgn)**.

- **Pre-Training Experiments with Two Attention Sinks (Appendix Section I)**: Reported on experiments with multiple sink tokens during the pre-training stage **(Reviewer MUgn)**.

These revisions and additions are aimed at comprehensively addressing the concerns raised by the reviewers and enhancing the understanding of StreamingLLM's capabilities and limitations. We thank the reviewers again for their thorough evaluations and look forward to your further feedback. Please do not hesitate to reach out if you have any additional questions or require further clarification.

---

### Meta-Review · Area_Chair_vbiF · 2023-12-06

**Metareview:**

This paper proposed an efficient attention mechanism, which is a combination of the sliding window attention plus the "token sink", a special token in the initial position. The authors experimentally validate the effectiveness of the proposed approach on extensive experiments. In particular, the approach is efficient and can enable LLMs to handle tasks with very long context lengths.

All reviewers appreciate the writing quality of this paper and give positive ratings, with which the AC agree. I am happy to see this paper accepted at the conference.

**Justification For Why Not Higher Score:**

The paper did not give any theoretical insight into why the approach works. The approach is largely based on empirical observation.

**Justification For Why Not Lower Score:**

All reviewers appreciate the contribution of this paper.

---

### Decision · Program_Chairs · 2024-01-16

Accept (poster)